# Pre-Transplantation Assessment of BK Virus Serostatus: Significance, Current Methods, and Obstacles

**DOI:** 10.3390/v11100945

**Published:** 2019-10-14

**Authors:** Fatima Dakroub, Antoine Touzé, Haidar Akl, Etienne Brochot

**Affiliations:** 1Agents Infectieux, Résistance et Chimiothérapie Research Unit, EA 4294, Jules Verne University of Picardie, 80000 Amiens, France; fatimadakroub20@gmail.com; 2Laboratory of Cancer Biology and Molecular Immunology, Faculty of Sciences-I, Lebanese University, Hadath 21219, Lebanon; haidar.akl@ul.edu.lb; 3Infectiologie et Santé Publique “Biologie des Infections à polyomavirus” team, UMR INRA 1282, University of Tours, 37082 Tours, France; antoine.touze@univ-tours.fr; 4Department of Virology, Amiens University Medical Center, 80000 Amiens, France

**Keywords:** BK virus, serological technique, BK virus serology, kidney transplantation

## Abstract

The immunosuppression required for graft tolerance in kidney transplant patients can trigger latent BK polyomavirus (BKPyV) reactivation, and the infection can progress to nephropathy and graft rejection. It has been suggested that pre-transplantation BKPyV serostatus in donors and recipients is a predictive marker for post-transplantation BKPyV replication. The fact that research laboratories have used many different assay techniques to determine BKPyV serostatus complicates these data analysis. Even studies based on the same technique differed in their standard controls choice, the antigenic structure type used for detection, and the cut-off for seropositivity. Here, we review the different BKPyV VP1 antigens types used for detection and consider the various BKPyV serostatus assay techniques’ advantages and disadvantages. Lastly, we highlight the obstacles in the implementation of a consensual BKPyV serologic assay in clinics (e.g., the guidelines absence in this field).

## 1. Introduction

The best treatment for patients with end-stage renal disease is kidney transplantation. However, technically successful transplantations can be complicated by renal dysfunction episodes in the following months [1]. There are many reasons for this renal dysfunction: Failure to control opportunistic infections, the antiviral and immunosuppressant drugs’ nephrotoxicity, and both acute and chronic immune-mediated graft rejection. The guidelines on the kidney transplant recipients treatment suggest that the immune mediated graft rejection can be mitigated by intensive immunosuppressant treatment in the immediate post-transplantation period [2]. The immunosuppression required for the graft function maintenance increases the risk of viral infections in kidney recipients [3]. A common condition in immunosuppressed individuals is the BKPyV reactivation [4]. Even though we still lack specific anti-BKPyV treatments, there are no methods for reliably predicting the onset of BKPyV-associated infectious complications. However, it has been postulated that the kidney allograft is the infection source; consequently, the donor’s BKPyV seroreactivity may reflect the subsequent BKPyV load in the recipient. Conversely, the recipient’s seroreactivity reflects his/her immune status against BKPyV. Hence, BKPyV serostatus is a valuable tool for predicting the BKPyV-associated disease occurrence after transplantation [5]. Here, we review and compare the different assay techniques used to assess BKPyV seroreactivity. We also consider the clinical BKPyV infection management as a function of the patient’s BKPyV serostatus. Lastly, we discuss the obstacles in the routine BKPyV serostatus assessment in a clinical setting.

## 2. Virology and Epidemiology of BKPyV

The BKPyV is a *Polyomaviridae* family member. It is a non-enveloped virus with a diameter of 45 nm and a ~5 kb double-stranded DNA genome. Four major subtypes (I to IV) have been described, with subtype I being the most prevalent worldwide [6]. These subtypes have been further divided into subgroups, and it has been shown that genotypes I, II, III, and IV behave as fully distinct serotypes [7]. The viral capsid’s outer surface is composed of 72 VP1 pentamers arranged in a T = 7 d icosahedral structure stabilized by calcium cations and disulfide bonds. The viral proteins VP2 and VP3 reside at the capsid’s inner part. DNA binding is mediated by the VP1 N-terminal domain, which lies inside the virion. A copy of VP2 or VP3 interacts with a VP1 pentamer through hydrophobic interactions [8]. After a primary BKPyV infection (which usually occurs during childhood), the virus becomes latent in the kidneys and the urinary tract. It can be reactivated in an immunosuppression context, leading in many cases to the virus particles excretion in the urine. It has been reported however, that occasional BKV excretion in the urine was detected in healthy adults and children as well [9]. An important risk factor for manifesting polyomavirus renal graft infection after transplantation is high dose immunosuppressive therapy [10]. BKPyV can also induce other diseases in immunocompromised patients (e.g., hemorrhagic cystitis in bone marrow transplant recipients and in cyclophosphamide-treated cancer patients). The guidelines for these conditions recommend regular BKPyV replication monitoring and immunosuppressant dose adjustment for patients with high viral loads [2].

## 3. Immune Control of BKVPyV

Both humoral and cellular immune responses are involved in the BKPyV infection control. In a cohort of renal recipients with BKPyVAN, Hariharan et al. observed significantly higher BKPyV-specific antibody titers in subjects after BKPyVAN resolution as compared to titers at BKPyVAN diagnosis time. The authors suggest that BKPyV-specific antibody titers are associated with viral clearance [11], but the increase in IgG levels can also be linked to viral replication. It has been suggested that BKPyV-specific T-cells play a dual role in the BKPyV infection control in renal transplantation patients. The reduction in immunotherapy leads to the cellular immunity restoration, which may succeed in inhibiting the infection and preventing BKVAN. On the other hand, if it fails to achieve viral clearance, the T-cell mediated immune response may add insult to the injury by homing T-cells to the graft and causing damage to the graft cells [12]. Only limited data exist on the innate immunity involvement in the BKPyV infection. However, a study by Womer et al. demonstrates lower levels of peripheral blood dendritic cells in patients with BKVAN compared to renal recipients with stable graft function. These cells are responsible for antigen presentation and T-cell activation [13].

## 4. BKPyV-Associated Nephropathy

Post-transplantation immunosuppression may lead to tBKPyV replication reactivation, which in turn may result in BK virus-associated nephropathy (BKPyVAN). This disease is a major renal allograft dysfunction cause (with a 1–10% incidence in kidney transplant patients) and can sometimes progress to interstitial nephritis with ureteric stricture and stenosis [14]. Several candidate biomarkers for BKPyV replication have been identified, such as decoy cells detection in the urine and BKPyV DNA load in urine and plasma [15]. Although BKPyVAN can appear as early as 1 month after transplantation, some cases are not detected until more than 80 months after the procedure. The viral reactivation is asymptomatic, and the infection is often only revealed by kidney failure. Despite a significant increase in clinical awareness and a better understanding of BKPyV infections, BKPyVAN still poses a real problem for kidney transplant patients [16]. The only available treatment strategy for BKPyVAN seeks to reduce virus replication while avoiding graft rejection; this corresponds to a timely level reduction of immunosuppression and (in some cases) antiviral therapy initiation [14]. Although a partial immune function restoration controls BKPyV replication, it increases the risk of the allograft immune rejection. There is a real need for controlled studies to find safe and effective treatment for BKVAN, especially for those in whom immunosuppression reduction is not possible [17].

## 5. Current Clinical Approaches for Assessing BKPyV Serology

Two risk factors for early post-transplantation BKPyV replication have been identified: A low BKPyV antibody titer in the recipient, and a high titer in the donor [18]. It has therefore been hypothesized that a single BKPyV serostatus assessment before transplantation can predict the post-transplantation BKPyV replication risk [5]. Despite these findings, a standardized, commercially available, regulatory-agency-approved assay for anti-BKPyV antibodies is not available [19]. More sensitive, standardized immunoassays would facilitate the donor/recipient immune status assessment and thus enable the clinician to more closely monitor patients with a high predicted viral replication risk [20]. Around the world, thousands of patients are on organ transplant waiting lists, and transplantation is becoming a major financial burden in the developed world [21]. Consequently, it is essential to improve BKPyV serologic assays and donor–recipient BKPyV seroreactivity matching with a view to increasing the kidney graft survival rate. To achieve this objective, the most cost-effective strategies for BKPyV screening in different patient populations must be determined—as noted in the Kidney Disease: Improving Global Outcomes guidelines [2].

Most serological assays detect antibodies against the immunodominant BKPyV capsid protein VP1; including enzyme-linked immunosorbent assays (ELISAs), neutralization assays, multiplex immunoassays, and hemagglutination inhibition assays. Most serologic assays detect the immunodominant BKPyV capsid protein VP1 (the virus’s major surface protein) [22]. Cost-effective strategies for BKPyV screening have been sought in various patient populations [23]. It is known that systemic BKPyV infections induce strong, stable, prolonged antibody responses against viral structural proteins. Thus, past BKPyV infections can be detected with high sensitivity by measuring the anti-VP1 antibodies accumulation. In contrast, antibodies against the large T-antigen (LT) are infrequent and have low titers—making them unsuitable infection markers in most cases [24]. The low antibody response against LT might be due to poor immune accessibility and/or poor recognition; the latter is thought to be due to the similarity between the LT functional domains and that of cellular proteins [25].

### 5.1. VP1 Antigens Used in BKPyV Serologic Assays

Although all serologic assays reviewed here detect anti-BKPyV VP1 antibodies, they differ regarding the target antigens. Furthermore, several different VP1 antigen types can be detected. Below, we briefly describe the VP1 antigens that have been incorporated into the serologic assays developed by research laboratories.

#### 5.1.1. Virus-Like Particles

Virus-like particles (VLPs) are most commonly generated from VP1 structural proteins, but VLPs with both VP1 and VP2/VP3 proteins have been synthesized. Although VLPs resemble native virions assembled into capsids (comprising 72 capsomers with a T = 7 symmetry), they do not contain viral genetic material. They can be used for diagnostic antigens for detecting serum specific antibodies against BKPyV VP1. The VLPs’ structure, transduction efficiency, and tropism are similar to those of native virions, except for the fact that VLPs do not undergo post-translational modification [26]. BK virus VLPs can encapsidate DNA fragments derived from the cells in which they were produced; consequently, the VLPs in each production batch contain VLPs with differing densities, depending on the incorporated DNA amount and size [27]. The VLPs quantity and quality can be affected by many factors, including the used production system type and the purification method. Virus-like particles can be produced in insect cells giving them the advantage of being free of mammalian pathogens; however, the yields are rather low, with a high cost and risk of contamination with enveloped baculovirus particles and host DNA [28]. Yeast production systems have the advantage of producing safe, DNA-free VLPs, which makes them perfect to produce VLP vaccines. In fact, a study found that recombinant VLPs synthesized in yeast and used in an ELISA for human polyomaviruses have many advantages in ease of production, protein yield, and cost terms [26]. The 293TT mammalian cell line is most commonly used for VP1 VLPs synthesis because it allows authentic assembly and folding of recombinant proteins. Still, the production costs in the mammalian system are high, yields are low, and the cells are vulnerable to infection with mammalian pathogens [28]. One must also consider the VLPs’ purity and integrity prior to the use in immunoassays. In fact, VLPs can be coupled to biotin for use in ELISAs. VLPs can be treated with the EZ-Link Sulfo-NHS-LC biotinylation kit and then bound to streptavidin plates, after which a sample diluent is added. Kardas et al. reported that standard polyomavirus VP1 VLPs and biotinylated VLPs did not differ significantly with regard to assay variability at the population level [29]. The VLP profile may vary even when the same production, purification, and quantification methods are applied. It is important to assess each batch’s quality by ensuring that the VLPs’ hemagglutination activity and immunogenicity make them suitable for serologic assays [27]. After production, SDS-PAGE can be used to confirm that the VLP batch has a major protein band at ∼40 kDa, and thus can be qualified for use in ELISAs [30]. It is known that native VLPs and denatured VLPs have different antigenic epitopes; denatured VLPs react less efficiently with BKPyV-positive human serum. BK virus VLPs are stable at relatively high pH values, which enables them to be used in conventional ELISAs [27]. These VLPs are therefore the best tools for detecting BKPyV seroreactivity and have also been extremely valuable in BKPyV epidemiological studies.

#### 5.1.2. Pseudovirions and Native Virion

The term “pseudovirion” (PsV) is used to describe synthetic viruses produced by the plasmid transfection of genes encoding capsid proteins and artificial genetic material used as a reporter. Although PsVs are similar to native virions in many ways (e.g., their behavior within cells), these synthetic viruses cannot replicate and do not propagate infection in cell cultures or in vivo. Hence, PsVs have become common tools for studying cellular entry and neutralization, and might be valuable in the future as vaccine vehicles or gene transfer tools [31]. Pastrana et al. generated pseudovirions by co-transfecting BKPyV capsid protein expression plasmids coding for VP1, VP2, and VP3 with a reporter plasmid encoding luciferase into 293TT cells. The cells were suspended and lysed 48 h post-transfection. The lysate was incubated overnight to allow capsid maturation, and then clarified. Ultracentrifugation using an iodixanol gradient was then used to purify the pseudovirions from the clarified supernatant [32]. Pseudovirions are mainly used in serum neutralization assays, where they contain a luciferase or green fluorescent protein reporter plasmid [33].

Apart from PsVs and VLPs, native virus particles can also be used as antigens in immunoassays. Native BKPyV particles are usually grown in HEK, Vero, or 293TT cells, harvested, purified, and quantified prior to their use in serologic assays [34]. It is also noteworthy that only the subtype Ia BKPyV (Dunlop or Gardner strain) can be propagated easily in culture, which means that the use of whole-virion antigens is not practical when the measurement of antibodies against different BKPyV serotype strains is required.

#### 5.1.3. Soluble VP1 Proteins

Both recombinant and synthetic soluble VP1 proteins have been used as antigens in ELISAs. In a computer-assisted analysis of the late viral region, Pirtrobon et al. produced two specific, synthetic BKPyV VP1 peptides with a stable secondary structure. The synthetic peptides were incorporated in ELISAs that could detect anti-BKPyV antibodies in the absence of cross-reactivity with other small DNA tumor viruses [35]. The use of uniform, well-defined synthetic peptides with a high epitope density advantageously limits inter- and intra-assay variability and increases sensitivity. However, cross-reactivity can still be a problem, since synthetic peptides may not be able to bind specifically enough to the target antibodies; the short peptides may have a different conformation when compared with full-length VP1 molecules assembled into capsomers during VLP synthesis [36]. Transfecting E.coli with pGEX VP1 plasmids produced VP1 pentamers; the resulting VP1 protein is fused to an N-terminal glutathione S-transferase (GST). After affinity purification on glutathione resin, the fusion proteins can be bound to 96-well polysorp plates (using a casein-glutathione conjugate) and used in a capture ELISA [37]. Alternatively, the VP1-GST fusion proteins can be directly affinity-purified on polystyrene beads for use in a multiplex immunoassay [24].

### 5.2. Assay Techniques for BKPyV Seroreactivity

Four different techniques can be used to evaluate seroreactivity to BK virus. The techniques’ respective advantages and disadvantages are summarized in Table 1.

#### 5.2.1. Enzyme Immunoassays

Enzyme-linked immunosorption is a rapid, high-throughput, sensitive, and highly reproducible method for antibody detection. Furthermore, colorimetric, chemiluminescent, or fluorescence ELISAs typically have a broad dynamic range [34]. The ELISA plates can be coated with any of the above-mentioned BKPyV antigens’ types. Kean et al. studied several human polyomaviruses and found that a VP1 pentamer-based ELISA performed better than the more common VLP-based ELISA. The casein-glutathione conjugate used to capture the GST-VP1 capsomers on Polysorp 96-well plates fully exposed the bound capsomers to the serum sample and facilitated all the VP1-reactive antibodies measurement [37]. However, Bodaghi et al. reported that ELISAs with VP1 VLPs as coating antigens are more specific and sensitive than those with VP1 monomers or pentamers. Furthermore, the researchers’ denaturation experiments experimentally confirmed the antigen’s three-dimensional structure importance [25]. In the absence of standardized ELISAs for BKPyV, research and clinical laboratories have developed their own in-house ELISAs using various antigens, protocols, and standards. This complicated the comparison of BKPyV ELISA serology results between one lab and another, especially in the absence of guidelines on quantitative cut-offs. As mentioned above, the BKPyV VP1 antigen can be used in different forms. Even labs that use the same type of antigen (VLPs, for example) can differ regarding the antigen production and purification methods and the final concentration used to coat wells. Another variable is the reference material used to optimize the assay, which may differ from one lab to another. For example, the negative control is a blank well in some studies [29,38] and a bovine serum albumin-coated well in others [39,40]. Bodaghi et al. used an SF9 extract as a negative control [25], while Abend et al. used human anti-chicken lysozyme IgG [23]. Similarly, the normalization well composition may vary, and some labs even skip this step. Inter-plate normalization usually involves diluting an internal reference serum close to 1 optical density. Hence, the absence of a standardized, commercially available antibody prevents labs from using the same identical normalization step. In addition to technical variables, the cut-off for positivity can be set differently in each laboratory. A clear BKPyV seropositive sample definition is currently lacking, and each laboratory uses its own in-house method to determine the cut-off. In summary, inter-ELISA variability is caused by differences in the reference material (normalization antibodies and negative controls), the VP1 antigen’s type and concentration, the experimental protocol, the cut-off, and the seropositivity definition. A growing body of evidence suggests that pre-transplantation testing for BKPyV exposure can help to predict the occurrence of BKPyV-associated diseases after transplantation. Despite that, there are currently no consensus guidelines on an ELISA technique that healthcare institutions could use to determine the BKPyV serostatus of kidney or bone marrow transplant recipients. It will be difficult (but not impossible) to implement a technique that can be universally applied for pre-transplantation BKPyV serology assessments in a clinical setting. In the light of research performed over the last decade (i.e., strong evidence of a relationship between pre-transplantation BKPyV serology and post-transplantation BKPyVAN), it is now more important than ever to develop a standard ELISA for pre-transplantation BKPyV serostatus.

#### 5.2.2. Multiplex Immunoassays

ELISA and other conventional serologic assays measure the presence of serum antibodies against a single antigen per well. In contrast, multiplex technologies enable the production of arrays of sensors—each of which provides its own unique detection signal. Multiple antigens can be measured simply by placing the sample in contact with the array [41]. Protein–protein interactions have been explored in multiplexed planar and suspension arrays, both of which requiring pre-purified proteins [42]. In a multiplex suspension array, a template (e.g., a micro well) is filled with different sensing elements in solution [41]. One of the best suspension array examples that efficiently detects antiviral antibodies in serum is the Luminex Multi-Analyte Profiling^®^ (xMAP^®^) technology, in which indicator molecules are covalently attached to 5.6-µm polystyrene bead sensor elements. The beads have an internal color code that is obtained by filling them with different proportions of two or three spectrally distinct fluorochromes—resulting in an array of at least 500 separate bead sets [42]. Thus, the difference in the internal classification dye quantity in each microsphere results in a unique emission profiles generation, even though these same-sized beads have similar emission requirements [43]. Luminex indirect serologic assays have been extensively validated for the detection of antibodies against several polyomaviruses types [44,45,46] including BKPyV [5,47,48]. The BKPyV VP1 protein was expressed in Escherichia coli as a fusion protein with GST, and then affinity-purified using Luminex beads coupled to glutathione-casein. The modified beads could to be used directly for the detection of anti-BKPyV antibodies [42]. When Luminex beads are used in serologic assays, non-specific background binding is a major drawback; human sera may contain antibodies that bind directly to the beads. Serum panels vary in the proportion of these sera, which frequently exceeds 5%. Using SeroMap beads (rather than xMAP^®^ beads) to minimize binding to heterophilic serum antibodies only partially solves the problem, so the sera pretreatment with background inhibitors was recently suggested [49]. Furthermore, seroepidemiologic studies require many samples to be tested for several analytes in a rapid, sensitive, specific manner. This kind of analysis is facilitated by multiplex assay formats. Hence, if one seeks to detect anti-BKPyV antibodies against several viral serotypes, multiplex technology will be a time saver. This technique allows the simultaneous analysis of each serum sample against all the BKPyV serotypes at once. Furthermore, multiplex technology minimizes the experimental variability associated with conventional serology methods because multiple data points are obtained from a single measurement. The technique’s requirement for a very low sample volume also maximizes data collection. In contrast, multiplex technology may offer fewer advantages in a clinical setting; the costly, specialized equipment and analytical software are unlikely to be available in all hospital laboratories. Compared with epidemiological studies, the number of subjects to be assessed at a given time point in a hospital or a transplant center is much lower. This means that the cost of performing these assays will be higher than for conventional serologic tests (e.g., ELISAs). Furthermore, it is harder to define a clear cut-off in multiplex assays, since the result for each sample is usually expressed as mean fluorescence intensity. Lastly, it is noteworthy that multiplex assays use soluble VP1 proteins (rather than VLPs); this may constitute a slight drawback because many studies have suggested that the conformational structure of VP1 inside VLPs offers more specificity and sensitivity than that of VP1 monomers or capsomers.

#### 5.2.3. Neutralization Inhibition Assays

A neutralization inhibition assay for BKPyV serology has been reported in the literature. In general, serum samples are serially diluted, pre-incubated with PsVs or native virions, added to seeded cells, and then incubated for a period of at least 48 or 72 h. The cell lysate is then analyzed: The greater the neutralizing antibodies titer in the serum is, the lower is the PsV-transduced or virion-infected cells’ number and thus the weaker is the signal [33]. Solis et al. synthesized three different PsV types and then measured the antibody titers against BKPyV in the sera of 156 kidney transplant recipients at six different time points. The researchers demonstrated that this technique could quantify antibody titers in many samples [50]. This technique’s greatest drawback is probably the need for cell culture—making it time-consuming, technically demanding and therefore unsuitable for clinical measurements. Furthermore, there is no standard method for a reliable neutralization inhibition assay so far, and as in the ELISA case, the seropositivity definition differs from one lab to another. Other variables include the BKPyV antigen type and the cell type used in the assay. For instance, RPTEC [23] and 293TT cells [7,32] have both been used to determine BKPyV serostatus.

#### 5.2.4. Hemagglutination Inhibition Assays

Many laboratories have used hemagglutination inhibition assays (HIAs) to measure the antibody titers to BKPyV because of the rapidity and ease with which they can be performed. However, HIAs are less sensitive and less accurate than enzyme immunoassays. Experiments with the HIA have shown that greatly differing anti-BKPyV titers and anti-JCV antibodies were obtained in individual sera, thus overcoming the cross-reactivity problem expected for JCV and BKPyV [34]. It is noteworthy that the HIA is technically demanding and cannot differentiate between different BKPyV serotypes.

### 5.3. Clinical Studies of BKPyV Serology

In Table 2, we provide an overview of the research studies conducted on BKPyV serostatus in kidney transplant donors and recipients. In 2017, Wunderink et al. established that donor pre-transplant BKPyV seroreactivity best predicted the occurrence of a manifest BKPyV infection in renal allograft recipients. The researchers found a strong correlation between donor BKPyV serostatus on the one hand and the development of post-transplantation BKPyV viremia and BKPyVAN on the other. These findings strongly suggest that the kidney allograft has an important role in the BKPyVAN development, since it acts as a vector for transmitting BKPyV to the recipient. Consequently, it is assumed that the intensity of the donor’s BKPyV seroreactivity corresponds to the infectious BKPyV load in the kidney allograft, which in turn is correlated with the BKPyV infection risk in the recipient. In contrast, the recipient’s BKPyV seroreactivity might reflect his/her overall anti-BKPyV immunity status. Thus, it may be relevant to assess the post-transplantation BKPyV infection risk by assaying for anti-BKPyV IgGs prior to kidney transplantation [5]. Similarly, Solis et al. found that patients who received a kidney graft from donors with elevated BKPyV-neutralizing antibody titers became positive for BKPyV DNA. The researchers also found that the recipient’s pre-transplantation titer of neutralizing antibodies against donor-specific BKPyV strains determined the BKPyV replication risk. Solis et al. suggested that physicians must take account the individual BKPyV risks when choosing immunosuppression strategies and monitoring patients after transplantation. Along with the recipient’s BKPyV DNA load, the neutralizing antibodies titer against the replicating strain is a valuable disease progression marker [50]. Similarly, many studies found that a positive donor BKV serostatus is associated with post-transplantation BKV infection [51,52,53]. In contrast, Abend et al. reported that BKPyV viremia was not significantly correlated with the recipient’s serostatus. This might have been because the anti-BKPyV antibodies levels were too low to provide protection in a transplantation context (i.e., with suppressed cellular immunity and elevated viral loads). Abend et al. suggested that BKPyV viremia may be due to a donor-virus-derived infection, and thus that it may be possible to identify recipients at a clinical BKPyV infection risk by measuring the donor’s serostatus [23]. On the other hand, Hirsch et al. proposed that the high-risk group to develop BKV infection after transplantation is not the seropositive donor and seronegative recipient transplant combination [15]. In view of these findings, we call on the scientific community to strive to (i) develop clear guidelines for assessing BKPyV serostatus, (ii) define quantitative cut-offs, and (iii) develop standard assay controls and reference samples. This will be the first step on the road to faithfully analyzing, comparing, and exploiting data on BKPyV serostatus and, ultimately, implementing these findings in clinical practice.

## Figures and Tables

**Table 1 viruses-11-00945-t001:** Advantages and disadvantages of assay techniques for BK polyomavirus (BKPyV) seroreactivity.

Technique Requirement	Advantages	Disadvantages	Time
Enzyme Immunoassay	Small quantities of sample required Versatile and customizable Inexpensive once set up	Can only measure one analyte at a time Cross-reactivity Relatively expensive initial investment Time consuming Elevated risk of error when testing a large number of samples	2 days (if wells are coated with antigen overnight)
Hemagglutination Inhibition Assay	Highly specific	Technically demanding Cannot distinguish between antibody classes Requires either intact virions or VLPs	1 day
Multiplex Assay	Simultaneous detection of multiple antigens High speed and dynamic range Customizable Reduced workflow	Expensive especially if a small number of antigens is analyzed Specialized equipment and analysis software are not available in most clinical settings Lack of normalization	Around 2 days (if beads are prepared in advance)
Neutralization Inhibition Assay	Highly Specific Measures neutralizing antibodies	Can only be used with PsV or viruses that can be grown Technically demanding Very time-consuming	Around 5 days

**Table 2 viruses-11-00945-t002:** An overview of the different research studies pertaining to the involvement of pre-transplant BKPyV serology testing in post-transplantation BKPyV infection.

Authors	Year	Number of Patients	Type of Assay and BKV Antigen	Conclusions from the Study
Solis et al. [50]	2018	168 KTR + 69 donors	Neutralization assay using pseudovirion system (BKPyV genotypes I, II, and IV)	Recipients with high neutralizing antibody titer have a lower risk for developing BKPyV viremia
Abend et al. [23]	2016	116 donor-recipient pairs	Neutralization inhibition assay using BKPyV particles (serotypes I, II, III, and IV) VLP-based ELISA to detect antibodies against BKPyV serotype I	Donor with significant serum neutralizing activity is associated with elevated risk for BKPyV viremia regardless of recipient serostatus
Wunderink et al. [5]	2016	407 donor-recipient pairs	Luminex assay detecting IgG reactivity against BKPyV Ib1 VP1 protein. *n* = 396 reanalyzed by VP1 VLPs-based ELISA to detect antibodies against BKPyV genotype Ib2	Donor BKPyV IgG levels were strongly associated with the occurrence of recipient viremia and BKPyVAN
Sood et al. [54]	2013	192 adult and 11 pediatric donor-recipient pairs	BKPyV VLPs-based ELISA to detect human IgG Antibodies	Infection was highest in the Donor+/Recipient− group and lowest in the Donor−/Recipient− group
Ali et al. [52]	2011	36 pediatric KTRs + donors	BKPyV VP1 VLPs-based indirect ELISA to detect human IgG antibodies	Low BKPyV serostatus in children is associated with a high risk of post-transplantation BKPyV viremia, particularly in the context of donor with high BKPyV serostatus
Bijol et al. [55]	2010	45 pediatric KTRs	BKPyV VLPs-based ELISA to detect human IgG antibodies	Positive recipient BKPyV serostatus did not confer protection to BKV after transplantation
Bohl et al. [56]	2008	87 KTRs	BKPyV VP1 VLPs-based ELISA to detect human IgG Antibodies	Pre-transplant seropositivity did not protect against sustained BKPyV viremia but it might mitigate the severity of infection
Bohl et al. [51]	2005	142 recipients and 84 donors	BKPyV VP1 VLPs-based ELISA to detect human IgG Antibodies	BKPyV infection in the recipient was strongly associated with a positive BKPyV donor antibody status
Smith et al. [57]	2004	173 pediatric KTRs	BKPyV VP1 VLPs-based indirect ELISA to detect human IgG Antibodies	Recipient seronegativity for BKPyV was significantly associated with the development of BKPyVAN
Hirsch et al. [15]	2002	77 KTRs	Hemagglutination inhibition assay	The high-risk group is not the seropositive donor and seronegative recipient transplant combination
Flegstad et al. [58]	1991	10 KTRs	Neutralization inhibition assay Hemagglutination inhibition assay IgG, IgA, and IgM ELISA	Positive recipient BKPyV serostatus did not confer protection to BKPyV after transplantation Children with BK nephritis demonstrated lower pretransplant antibodies levels when compared to control groups (no infection)
Andrews et al. [53]	1988	496 donor-recipient pairs	Hemagglutination inhibition assay	A seropositive donor increased the rate of primary and reactivation infections with BKPyV

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
