# Peer review of "Pre-Transplantation Assessment of BK Virus Serostatus: Significance, Current Methods, and Obstacles"

_viruses, 2019, doi:10.3390/v11100945_

Round 1

Reviewer 1 Report

The review submitted by Dakroub et al focuses on BK polyomavirus serologic assays, comparing advantages and disadvantages of different methods.

In the previous years, there were an increasing number of publications studying the humoral response during BKPyV infection. It is difficult to compare their results due to the heterogeneity of the methods used. Comparing these methods and the results they provided is thus interesting. This review could be a helpful guide for interpretation of studies focusing on BKPyV humoral response, but many comments can be made on this manuscript, which requires major revision.

General comments

First of all, the English language has to be revised, a major part of the references are not at the right place, or not the good ones, and the form has to be improved. The author should check again the entire manuscript for the references. A

Specific Comments (not all the ref errors are listed)

Here are some examples of points to correct:

L42 “these factors” > if factors refer to the previous sentence, then not all of them are mitigated by intensive immunosuppressant treatment ! (ex opportunistic infections).

L42 ref 2 is not the required ref for international recommandations

L46 ref 4 refers to Stem cell transplantation

L58 BKPyV is to be used, instead of BK virus (or at least chose one of them). Polyomaviridae refers to a family, and has to be written in italic.

L62 ref 6 is not the good one.  In fact, serotypes were discovered first, then the 4 genotypes were found to correlate to these serotypes. The finding that Ib-1 and Ib-2 are different serotypes is an artefact (use of a mutated Ib-2 strain, not representative of Ib-2 WT).

L64a and 66, again not the good ref 7 and 8

L82 thanks to a systematic screening and modulation of immunosuppressive therapy, the percentage of graft loss due to PVAN has decreased

L89 ref 13 nothing to do with the sentence

L119 “due to sequence homology….and thus immunologic tolerance” Really? Could you explain this affirmation?

Paragraph 4.1.1 > the words used are rather vague. It would be useful to include a table with yields and possibly costs for the different methods of VLP production

L146 to 150 it is not clear whether this part refers to a specific publication

L153 why two major bands for VP1 VLPs?

Figure 1 : quality has to be improved. Presentation of the different methods should be in the same way as on the text (1A > ELISA, 1B > Multiplex, C> NIA, D < HIA). Regarding NIA, most of these methods are performed using pseudovirus, not infectious ones. it should be stated in the figure

Paragraph 4.1.2 description of pseudovirion production is not clear

L193  wrong. Dunlop is not the only strain that can be propagated in culture

Paragraph 4.1.3: L207 “recombinant soluble VP1 proteins” > pentamers maybe??

L223 pentamer instead of capsomer

L286-287 “up to thousand serum samples”. Not sure. Luminex allows to analyse serological responses to multiple antigens in a single assay, not multiple sera in a single assay

Paragraph 4.3

All of this section is to be revised. It comes back on part of the introduction, gives a summary of 3 recent studies but ignores previous important publications. A table summarizing the major results of the publications would have been really helpful, comparing the number of patients, the nature of BKPyV antigens used, assay type, correlation with BKPyV infection….then the authors could reach some meaningful conclusions.

Author Response

Answers to Reviewers' Comments

We thank the reviewers for the constructive comments and for their interest in our manuscript. We have now carefully considered the remarks of both referees, and we have adapted the text or added extra comments, references, paragraphs, and tables as outlined below. We are very confident that we have responded to all remarks and we hope that you will find our manuscript now acceptable for publication. Please note that the paragraph and line numbering in this letter are now adapted from the corrected manuscript.

Reviewer 1 :

The review submitted by Dakroub et al focuses on BK polyomavirus serologic assays, comparing advantages and disadvantages of different methods. In the previous years, there were an increasing number of publications studying the humoral response during BKPyV infection. It is difficult to compare their results due to the heterogeneity of the methods used. Comparing these methods and the results they provided is thus interesting. This review could be a helpful guide for interpretation of studies focusing on BKPyV humoral response, but many comments can be made on this manuscript, which requires major revision.

Thank you for your comment.

General comments

First of all, the English language has to be revised, a major part of the references are not at the right place, or not the good ones, and the form has to be improved. The author should check again the entire manuscript for the references.

As you suggest, the English language has been revised by native English speakers and references have been checked one by one.

Specific comments :

L42 “these factors” > if factors refer to the previous sentence, then not all of them are mitigated by intensive immunosuppressant treatment ! (ex opportunistic infections).

We have now specified the ‘’immune mediated graft rejection’’ as the factor that can be mitigated by intensive immunosuppression.

L42 ref 2 is not the required ref for international recommandations

L46 ref 4 refers to Stem cell transplantation

L64a and 66, again not the good ref 7 and 8

L89 ref 13 nothing to do with the sentence

We have removed the above references and replaced them with relevant references of papers that specifically deal with the text content.

Reference 2 is replaced by that of the KDIGO guidlines for the care of kidney transplant recipients. (L44)

Reference 4 for was removed and replaced by that of a paper pertaining to renal transplantation instead of stem cell transplantation. (L50)

Reference 7 : data contained in the sentence referring to reference 7 was altered to include more details about BKV reactivation. The reference was removed and replaced by the correct one. (L76)

The assertation which was linked to reference 8 is now completely removed and replaced by a statement compatible with the new reference. (L81)

Ref 13 : we have changed the sentence to include a recommendation for controlled studies to find safe and effective treatment for BKVAN. The old reference has been removed and replaced by the reference from the new relevant paper. (L123)

L58 BKPyV is to be used, instead of BK virus (or at least chose one of them). Polyomaviridae refers to a family, and has to be written in italic.

We chose to use BKPyV and checked that all the manuscript now contains the mentioned abbreviation instead of BK virus. We also changed the word Polyomaviridae to the italic format.

L62 ref 6 is not the good one. In fact, serotypes were discovered first, then the 4 genotypes were found to correlate to these serotypes. The finding that Ib-1 and Ib-2 are different serotypes is an artefact (use of a mutated Ib-2 strain, not representative of Ib-2 WT).

The reference was removed and replaced by that of a paper by Pastrana et al on “BK Polyomavirus Genotypes Represent Distinct Serotypes with Distinct Entry Tropism”.

It has been indicated in L68 that subtypes I, II, III, and IV  behave as fully distinct serotypes (Ib2 was not included).

There is no hint in the text about the chronological order of serotype and genotype discovery.

L82 thanks to a systematic screening and modulation of immunosuppressive therapy, the percentage of graft loss due to PVAN has decreased

We removed the sentence indicating that 50% of patients with BKVAN will lose the graft within 6 months. (L115)

L119 “due to sequence homology….and thus immunologic tolerance” Really? Could you explain this affirmation?

In an attempt to explain the lower-antibody response against the large T antigen, Bodaghi et al hypothesized that the difference in immunogenicity between BKV VP1 and BKV LT could be due to insufficient recognition of LT by the immune system. The authors propose that the fact that LT contains functional domains with high sequence homology to cellular proteins, such as the ATPase and p53 binding domains, a certain degree of immunological tolerance may have been established (not complete tolerance)

We have changed the sentence in Line 158 to better reflect the above theory.

Paragraph 4.1.1 > the words used are rather vague. It would be useful to include a table with yields and possibly costs for the different methods of VLP production

Thank you for this comment which allowed us to make many changes in this paragraph rather than proposing an additional table.

L146 to 150 it is not clear whether this part refers to a specific publication

L186 to 190 describes the production of biotinylated VP1 VLPs and compares them to standard VP1 VLPs. Its reference is a publication by Kardas et al (number 29 in the reference list of the manuscript)

L153 why two major bands for VP1 VLPs?

In the materials and methods section of their paper, Harihan et al indicated that BKV VLP lots that demonstrated major protein bands migrating at 40 and 30 kDa qualified for use in ELISA.

Based on methodology from a different paper and data from UniProtKB (Mass BKV VP1= 40,109 kDa) we changed the sentence in L194 to indicate that a major band for VP1 VLPs is obtained at 40 kDa.

Figure 1 : quality has to be improved. Presentation of the different methods should be in the same way as on the text (1A > ELISA, 1B > Multiplex, C> NIA, D < HIA). Regarding NIA, most of these methods are performed using pseudovirus, not infectious ones. it should be stated in the figure

       Thank you for your comments. Since this figure is mentioned as not essential by the second reviewer, we decided to delete it and instead add a table listing the different clinical studies on BKPyV serology.

Paragraph 4.1.2 description of pseudovirion production is not clear

We added a brief description of PsV production adapted from the methods used by Pastrana et al.

L193 Dunlop is not the only strain that can be propagated in culture

              We adjusted this statement to include the gardner strain along with dunlop.

              We intended to inform that only the Ia subtype is propagated in culture with good feasibility.

Paragraph 4.1.3: L207 “recombinant soluble VP1 proteins” > pentamers maybe??

“recombinant soluble VP1 proteins” was removed and replaced by “VP1 Pentamers’’ in L257

L223 pentamer instead of capsomer

“capsomer’’ in L273 was replaced by “pentamer’’

L286-287 “up to thousand serum samples”. Not sure. Luminex allows to analyse serological responses to multiple antigens in a single assay, not multiple sera in a single assay

We have changed the sentence in L339 to indicate that the multiplex assay allows the simultaneous analysis of each serum sample against all the BKPyV serotypes at once.

Paragraph 4.3 : All of this section is to be revised. It comes back on part of the introduction, gives a summary of 3 recent studies but ignores previous important publications. A table summarizing the major results of the publications would have been really helpful, comparing the number of patients, the nature of BKPyV antigens used, assay type, correlation with BKPyV infection….then the authors could reach some meaningful conclusions.

We appreciated this comment from the referee, and we have now added to paragraph 5.3 previous representative and important papers on BKPyV serology and BKPyV infection. We also included a table summarizing the major results of the publications as suggested by both of the reviewers.

Reviewer 2 Report

The review provided by Dakroub, Touze, Akl and Brochot touches a very important issue: to standardize methods in BKV serostatus determination. The authors describe the findings of several studies analyzing the BKV serostatus pre-transplantation to make risk estimations on BKV re-activation in the recipient. 

By providing an overview on BKV biology in patients and the current methods used to answer this question, the authors reach out to the scientific community to develop guidelines, standard assay controls to implement BKV serostatus into clinical practice. 

major criticism:

Figure 1 if needed at all (the information of this figure can be integrated into Table 1) would need graphical improvement. In my opinion, instead of Figure 1 it would be more informative to the reader to include a table in which all these different clinical studies would be listed (with their advantages and disadvantages) The paragraph on Virology and epidemiology of BKV is rather short and would profit by including more specific information on the immune control of BKV

minor points:

line 89: reference 13 might be not correct line 111-112: should reads: most serological assays detect antibodies against the immunodominant BKV capsid protein VP1 line 129: VLPs are not limited to VP1, they can consist of VP1, 2 and 3.

Author Response

Answers to Reviewers' Comments

We thank the reviewers for the constructive comments and for their interest in our manuscript. We have now carefully considered the remarks of both referees, and we have adapted the text or added extra comments, references, paragraphs, and tables as outlined below. We are very confident that we have responded to all remarks and we hope that you will find our manuscript now acceptable for publication. Please note that the paragraph and line numbering in this letter are now adapted from the corrected manuscript.

Reviewer 2 :

The review provided by Dakroub, Touze, Akl and Brochot touches a very important issue: to standardize methods in BKV serostatus determination. The authors describe the findings of several studies analyzing the BKV serostatus pre-transplantation to make risk estimations on BKV re-activation in the recipient.

By providing an overview on BKV biology in patients and the current methods used to answer this question, the authors reach out to the scientific community to develop guidelines, standard assay controls to implement BKV serostatus into clinical practice.

Thank you for your comment.

Major Criticism :

Figure 1 if needed at all (the information of this figure can be integrated into Table 1) would need graphical improvement.

       We have removed figure 1 from the manuscript

In my opinion, instead of Figure 1 it would be more informative to the reader to include a table in which all these different clinical studies would be listed (with their advantages and disadvantages)

We appreciated this comment from the referee, and we have now added a table summarizing the major results of the publications on BKPyV serology and its correlation with BKPyV infection as suggested by both of the reviewers.

The paragraph on Virology and epidemiology of BKV is rather short and would profit by including more specific information on the immune control of BKV

To the Virology and epidemiology paragraph of the manuscript, we have now added more information on the structure and protein content of BKPyV.

In addition, we included a new paragraph on the immune control of BKPyV infection (Page 2/ paragraph 3).

Minor points :

line 89: reference 13 might be not correct

We have changed the sentence to include a recommendation for controlled studies to find safe and effective treatment for BKVAN. The old reference has been removed and replaced by the reference from the new relevant paper.

line 111-112: should reads: most serological assays detect antibodies against the immunodominant BKV capsid protein VP1

We have changed the sentence in L146 to mention that most serological assays detect antibodies against the immunodominant BKV capsid protein VP1.

line 129: VLPs are not limited to VP1, they can consist of VP1, 2 and 3

Line 168 now indicates that Virus-like particles (VLPs) are most commonly generated from VP1 structural proteins, but VLPs with both VP1 and VP2/VP3 proteins have been synthesized.

Round 2

Reviewer 1 Report

The manuscript has been revised by the authors, and, except some minor corrections (in yellow in the attached pdf file), is now suitable for publication

Author Response

Answers to Reviewers’ Comments (Round 2)

Letter to the editor,

Dear Sir,

We thank the reviewers for the constructive comments and for their interest in our manuscript. We have now carefully considered the remarks of both referees, and we have adapted the text or added extra comments, references, paragraphs, and tables as outlined below. We are very confident that we have responded to all remarks and we hope that you will find our manuscript now acceptable for publication. Please note that the paragraph and line numbering in this letter are now adapted from the corrected manuscript.

Reviewer 1:

The manuscript has been revised by the authors, and, except some minor corrections (in yellow in the attached pdf file), is now suitable for publication

Thank you for your comments. They were very helpful to ensure a better quality of the manuscript.

English language editing:

We have removed the extra spaces, commas, or dots found in the manuscript, and made sure that all reference numbers are preceded by a space and not closely linked to the sentence before. Such modifications were not tracked in the new manuscript, but all the suggested changes by the reviewer were completed.

Line 80: really? I would be more cautious about that, these results have not been really reproduced and the increase in IgG levels seems to be linked to viral replication, but cell-mediated response is more clearly related to viremia clearance

The phrase at line 80 was modified to demonstrate the study’s finding without suggesting a role of anti-BKPyV antibodies in viral clearance. We also reflected the reviewer’s view that the mentioned antibodies can be linked to viral replication.

Line 168: the switch to the past is not appropriate here

We agree and we modified the sentence accordingly (Line 172).

To my opinion, the ref hariharan 2005, cited by the authors at the beginning of the manuscript, lacks here.

Table 2 only includes studies pertaining to BKPyV serology before transplantation. The study by Hariharan et al investigates the anti-BKPyV humoral response after transplantation, and therefore is not included in the mentioned table.

Reviewer 2 Report

The authors addressed all points raised by the reviewer and significantly improved the review.

I have no further comments.

Round 3

Reviewer 1 Report

No restriction for the publication of this revised manuscript